# NMR-Based Metabolomics to Analyze the Effects of a Series of Monoamine Oxidases-B Inhibitors on U251 Cells

**DOI:** 10.3390/biom13040600

**Published:** 2023-03-27

**Authors:** Zili Guo, Jinping Gu, Miao Zhang, Feng Su, Weike Su, Yuanyuan Xie

**Affiliations:** 1Key Laboratory for Green Pharmaceutical Technologies and Related Equipment of Ministry of Education, College of Pharmaceutical Sciences, Zhejiang University of Technology, Hangzhou 310006, China; 2Zhejiang Key Laboratory of Pollution Exposure and Health Intervention, Interdisciplinary Research Academy, Zhejiang Shuren University, Hangzhou 310015, China; 3Collaborative Innovation Center of Yangtze River Delta Region Green Pharmaceuticals, Zhejiang University of Technology, Hangzhou 310006, China

**Keywords:** NMR, metabolomics, metabolic pathway, Alzheimer’s disease, coumarin derivative

## Abstract

Alzheimer’s disease (AD) is a typical progressive neurodegenerative disorder, and with multiple possible pathogenesis. Among them, coumarin derivatives could be used as potential drugs as monoamine oxidase-B (MAO-B) inhibitors. Our lab has designed and synthesized coumarin derivatives based on MAO-B. In this study, we used nuclear magnetic resonance (NMR)-based metabolomics to accelerate the pharmacodynamic evaluation of candidate drugs for coumarin derivative research and development. We detailed alterations in the metabolic profiles of nerve cells with various coumarin derivatives. In total, we identified 58 metabolites and calculated their relative concentrations in U251 cells. In the meantime, the outcomes of multivariate statistical analysis showed that when twelve coumarin compounds were treated with U251cells, the metabolic phenotypes were distinct. In the treatment of different coumarin derivatives, there several metabolic pathways changed, including aminoacyl-tRNA biosynthesis, D-glutamine and D-glutamate metabolism, glycine, serine and threonine metabolism, taurine and hypotaurine metabolism, arginine biosynthesis, alanine, aspartate and glutamate metabolism, phenylalanine, tyrosine and tryptophan biosynthesis, glutathione metabolism and valine, leucine and isoleucine biosynthesis. Our work documented how our coumarin derivatives affected the metabolic phenotype of nerve cells in vitro. We believe that these NMR-based metabolomics might accelerate the process of drug research in vitro and in vivo.

## 1. Introduction

Metabolism is the central role of systems biology, and is also amplified relative to changes in the DNA, RNA and enzyme activities [1]. Since the metabolic network is interconnected, its proper operation requires a number of regulators to maintain regulatory stability. These cellular regulators include transcription factors [2], isomers of metabolites [3], signal transduction molecules [4] and immune factor [5]. Thus, there is a definite need for a tool for medication development to monitor changes in metabolism.

For accelerating drug discovery, increased techniques are being used for high-throughput screening. These techniques mainly contain silico tests [6] and biochemical [7], genetic [8] and pharmacological tests [9]. However, most tests monitor a single signal which relates to a single indicator. On the other hand, as high through technology matures, multi-index notation could be observed synchronously. Genomics, transcriptomics, proteomics and metabolomics were used in drug research and development [10,11,12,13]. These omics technologies could provide multiple data more clearly linked to the drug targets. Nuclear magnetic resonance (NMR)-based metabolomics could be an appropriate tool for analyzing the changes in metabolic phenotypes. Tiziani et al. used NMR-based metabolomics in cell culture medium for screening a series of kinase inhibitors and found the relationship between pharmacodynamics and metabolism [14]. In our earlier investigations, the toxicity mechanism of chemicals in in vitro cells was assessed using NMR-based metabolomics [15,16].

Alzheimer’s disease (AD) is a typical progressive neurodegenerative disorder accounting for up to 75% of all dementia cases [17]. There are multiple possible pathogeneses, including β-amyloid (Aβ) deposits [18], hyper-phosphorylated π-protein aggregation [19], oxidative stress and low levels of acetylcholine (Ach) [20]. In the last few years, monoamine oxidases (MAOs) have been paid more attention for their potential use in treating AD; they are important enzymes in monoamine neurotransmitter metabolism [21,22]. MAO-A could degrade the biogenic amine serotonin and regulate neurotransmitter metabolism [23]. Research has shown that the expression of MAO-B increased with age, and activity of MAO-B could increase neuronal damage [24]. Several research teams have investigated the synthesis, biological assessment, and computational aspects of cholinesterase and MAO inhibitors [25,26,27,28]. Potential MAO-B inhibitors could also be screened from natural products of roots of Platycodon grandiflorus (Jacq.) A.DC. [29]. Holger Stark et al. found that neuroprotectant ASS234 could inhibit the neurotransmitter-catabolizing enzymes (ChEs and MAOs) alongside H3R affinity for neurodegenerative diseases, including AD and Parkinson’s disease [30]. Albreht et al. explained the mechanism of irreversible MAO inhibitors by using the covalent cyanine structure linking the multi-target propargylamine inhibitor ASS234 and the flavin adenine dinucleotide in MAO-A [31]. Coumarin derivatives could also be used as potential inhibitors [32,33]. Nicola et al. derived a predictive three-dimensional structure of the target molecules (3D-QSAR) model employed for guiding the rational design of 10 new potent and selective MAO B inhibitors [34]. Analogously, we synthesized a series of coumarin derivatives as potential therapeutic AD drugs, which have effects as multifunctional brain permeable iron chelators and MAO-B inhibitors. In this work, we used NMR-based metabolomics for the pharmacodynamic appraising of the potentially active compounds. Then, we described the changes in metabolic profiles of nerve cells with different coumarin derivatives by using NMR-based metabolomics. This work would be extremely valuable for accelerating the pharmacodynamic assessment of potential compounds for the research and development of drugs.

## 2. Materials and Methods

### 2.1. Chemical Reagents and Cell Culture

NaH_2_PO_4_·2H_2_O and K_2_HPO_4_·3H_2_O (analytical grade) were purchased from J&K Scientific Ltd. (Shanghai, China). D_2_O (purity 99.9%) and sodium 3-(trimethylsilyl) propionate-2,2,3,3-d4 (TMSP) were bought from Cambridge Isotope Laboratories, Inc. (Tewksbury, MA, USA). The twelve coumarin derivatives were synthesized and purified in our lab. The detailed synthesis steps have been published previously [35], and the details are listed in the Appendix A. These twelve coumarin derivatives were dissolved in dimethyl sulfoxide (DMSO), which was obtained from Sigma-Aldrich (St. Louis, MO, USA). The water used in the experiments was purified using a Milli-Q system (Merck KGaA, Darmstadt, Germany). The astrocytoma cell line U251 was purchased from the China Center for Typical Culture Collection (CCTCC). Then, the cells were cultured in Dulbecco’s Modified Eagle Medium (DMEM, HyClone^TM^ Cytiva, Logan, UT, USA) with two antibiotics (penicillin, 100 units/mL; streptomycin, 100 μg/mL) and 10% fetal bovine serum (FBS, HyClone^TM^ Cytiva, Logan, UT, USA) in an incubator (5% CO_2_; 37 °C).

### 2.2. MTS Cell Proliferation Assay

The MTS kit (MTS assay, Promega, Madison, WI, USA) was used to assess the cell viability for calculating the half-maximal inhibitory concentration (IC_50_) of these coumarin derivatives. The U251 cell was cultured in 96-well plates (about 5 × 10^3^/well). Then, the cells were incubated with 200 μL supernatant and treated with various concentrations of coumarin derivatives (20, 40, 60, 80, 100, 120 μM). After 24 h of incubation, cell samples were then co-cultured with MTS kit (20 μL/well) and culture medium (DMEM, 100 μL/well) for 4 h. Next, colored MTS products were detected in the Microplate Reader (Spark^®^, Tecan Inc., Männedorf, Switzerland) by using the absorbance of each well at 490 nm. A detailed cell proliferation assay was operated as previously described [36]. Then, we chose the tenth of IC_50_ of each coumarin derivative for the following experiments.

### 2.3. Cell Intracellular Extracts’ Preparation and NMR Measurements

Before metabolite extraction, about 1 × 10^7^ cells were seeded in 10 cm diameter culture dishes and incubated for 24 h at 37 °C and 5% CO_2_. The cells were then harvested and quenched with a direct cell quenching method by using the frozen methanol [37]. The cell intracellular metabolites were extracted by using mixed solution system (CHCl_3_/CH_3_OH/H_2_O), and the detailed steps were described as in a previous article [15,36]. The mixed solution system was layered after centrifugation, and intracellular aqueous metabolites were in the upper (CH_3_OH/H_2_O). The nitrogen blowing concentrator was applied to remove the upper solvents. Then, the aqueous metabolites were re-dissolved in 500 μL phosphate buffer (150 mM K_2_HPO_4_/NaH_2_PO_4_, 0.5 mM TMSP, 10% D_2_O, 1‰ NaN_3_). After centrifugation, the supernatant was transferred to 5 mm NMR tube. All samples were analyzed in the BRUKER AVANCE III HD 600 MHz spectrometer (BRUKER BioSpin, Germany). The one-dimensional ^1^H spectra were operated in the TXI probe at 298 K. Using the NOESYPR1D [RD-90°-t_1_-90°-τm-90°-acq] pulse sequence, the water suppression irradiation was led to the supersaturation stage with 3 s relaxation delay and 120 ms mixing time. The detailed acquisition parameters were described in our previous works [15,36].

### 2.4. Identification of Metabolites and Statistical Analysis

The Chenomx NMR Suit (Version 7.1, Chenomx Inc., Edmonton, AB, Canada) was applied for identifying the metabolites from the NMR spectra of samples. Then, we used the HMDB database (URL: http://www.hmdb.ca/; access on 17 August 2022) to verify the metabolite NMR spectrum. These targeted metabolites were quantified from complex ^1^H NMR spectra by using an Automated Quantification Algorithm (AQuA) [38]. The principal component analysis (PCA) [39] and hierarchical clustering analysis (HCA) [40] were performed to show clusters among all samples. Then, orthogonal partial least squares-discrimination analysis (OPLS-DA) was applied for distinguishing the metabolic phenotypes between the treatment groups and control group [39], and the corresponding response permutation testing (RPT) was used for verifying the robustness of OPLS-DA models [41]. In the OPLS-DA models, the variable importance in the projection (VIP) [42] and the correlation coefficients (r) for the variables that are related to the first predictive component (tp1) [43] were used for the differential selection of metabolites. In addition, the probability *p* values by CV-ANOVA and fold changes were also calculated between treatment groups and the control group for assessing the statistical significance of differential metabolites. These four parameters were used in the enhanced volcano plots for visualizing the differential metabolites [41,44].

### 2.5. Disturbed Metabolic Pathway Analysis

Based on the different metabolites, metabolic pathway analysis was used to identify significantly disturbed pathways associated with the treatment groups. These works were performed on the Pathway Analysis module from the web-sever of MetaboAnalyst 4.0 (URL: www.metaboanalyst.ca/; access on 17 August 2022) [45].

## 3. Results

### 3.1. Effect of Coumarin Derivatives on Cell Viability

In the current research, the results of the MTS Assay showed that the range of IC_50_ of these twelve coumarin derivatives was 50–100 μM (Appendix A). These data indicated that these twelve compounds had low toxicity. For studying the effect of coumarin derivatives on cell metabolism with non-cytotoxicity, we chose the tenth of IC_50_ of each coumarin derivative for the subsequent studies.

### 3.2. Metabolic Profiles Analysis of U251 Cells in Twelve Coumarin Derivatives

Using the BMRB database (URL: http://www.bmrb.wisc.edu/, access on 17 August 2022), HMDB database (URL: https://hmdb.ca/; access on 17 August 2022) and Chenomx NMR Suite for assigning the metabolites from the NMR spectra, we identified 58 metabolites for quantitative calculation in MATLAB (Version 2015b, MathWorks, Inc., Natick, MA, USA) with AQuA (Figure 1 and Appendix A). Then, multivariate statistical analysis was used to analyze the quantitative data of the metabolites. In the PCA scores plots, metabolic profiles of cell lines in the treatment with different coumarin derivatives were different (Appendix A). Meanwhile, the result of HCA also displayed that the metabolic phenotypes were different (Appendix A). Then, the supervised multivariate statistical analysis was also applied to distinguish the metabolic profiles. The PLS-DA scores plots and corresponding RPTs indicated that the metabolic profiles of cell lines in the treatment with coumarin derivatives could be distinguishable from the control group (Appendix A).

### 3.3. Determination of Differential Metabolites in the Treatment with Different Coumarin Derivatives

For analyzing the differential metabolites, the four dimensional enhanced volcano plots were used for data visualization [46]. The VIP values and correlation coefficients (r) in the OPLS-DA models were calculated in the SMICA-P14+. The scores plots of OPLS-DA models also showed that the metabolic profiles of cell lines in the treatment of different coumarin derivatives were differentiable (Appendix A). In the enhanced volcano plot (Figure 2 and Figure 3), the differential metabolites were determined using the following four criteria: VIP value > 1; correlation coefficient (r) > 0.497; *p* value < 0.05; and the absolute log_2_ (fold change) > 0.25. The differential metabolites are located at the upper-left and upper-right areas of the volcano plot, with larger circle shapes and gradual warm colors. According to Figure 2A, the metabolites of glycine, guanidinoacetate and lysine were increased and the metabolites of aspartate, taurine and galactitol were decreased in the U251 cells co-incubated with CD1. In the U252 cells co-incubated with CD2 (Figure 2B), three metabolites were increased including glutamine, glutamate and galactitol, and three metabolites were decreased, including taurine, inosine and lysine. When the U251 cells were incubated in CD3, there were four metabolites increased, including phenylalanine, glutamine, glutamate and NAD+; there were three metabolites decreased, including taurine, aspartate and lysine (Figure 2C). Then, the differential metabolites were changed when the cells in the culture medium had added CD4; five metabolites were increased (galactitol, NAD+, glutamine, glutamate and glutathione), and two metabolites were decreased (taurine and lysine) (Figure 2D). When the cells in the culture medium had added CD5 (Figure 2E), three metabolites were increased (ADP, glutamine and glutathione) and two metabolites were decreased (taurine and lactate). In the culture medium with CD6 (Figure 2F), there were two increased metabolites (glycine and lactate) and three decreased metabolites (aspartate, taurine and glutathione). Similarly, we analyzed the differential metabolites in the U251 cells co-incubated with the subsequent six coumarin derivatives (Figure 3). Detailed statistical information for each group of differential metabolites is provided in Appendix A.

### 3.4. Significantly Disturbed Metabolic Pathways in the Treatment with Different Coumarin Derivatives

Based on the differential metabolites, we identified significantly disturbed metabolic pathways in the treatment with different coumarin derivatives (Figure 4 and Figure 5). For the U251 cell in the treatment with CD1, there were three significantly disturbed metabolic pathways including aminoacyl-tRNA biosynthesis, glycine, serine and threonine metabolism, and taurine and hypotaurine metabolism (Figure 4A). When the co-cultured compound changed CD2, the disturbed metabolic pathways also changed, including D-glutamine and D-glutamate metabolism, aminoacyl-tRNA biosynthesis, arginine biosynthesis, alanine, aspartate and glutamate metabolism, and taurine and hypotaurine metabolism (Figure 4B). When the U251 cell was co-cultured with CD3, the metabolic pathways of aminoacyl-tRNA biosynthesis, arginine biosynthesis, alanine, aspartate and glutamate metabolism, D-glutamine and D-glutamate metabolism and phenylalanine, tyrosine and tryptophan biosynthesis were disturbed (Figure 4C). The disturbed metabolic pathways between these two coumarin derivatives were similar. Compared with the previous coumarin derivatives (CD1, CD2 and CD3), the metabolic pathway of glutathione metabolism was an emerging perturbed metabolic pathway (Figure 4D). There were four metabolic pathways disturbed for the cell lines in the treatment of CD5, including D-glutamine and D-glutamate metabolism, arginine biosynthesis, alanine, aspartate and glutamate metabolism, and taurine and hypotaurine metabolism (Figure 4E). When the U251 cell was in the treatment with CD6, the disturbed metabolic pathways were changed including glutathione metabolism, aminoacyl-tRNA biosynthesis and taurine and hypotaurine metabolism (Figure 4F). The disturbed metabolic pathways of subsequent coumarin derivatives that interfere with cells are basically the same as those described above. Of course, there were also some derivatives that cause changes in other metabolic pathways. There were five disturbed metabolic pathways for the U251 cell in the treatment with CD7, including D-glutamine and D-glutamate metabolism, arginine biosynthesis, glutathione metabolism, alanine, aspartate and glutamate metabolism, and taurine and hypotaurine metabolism (Figure 5A). The disturbed metabolic pathways of arginine biosynthesis, alanine, aspartate and glutamate metabolism and phenylalanine, lysine and tryptophan biosynthesis were changed when the cells were co-cultured with CD8 (Figure 5B). The metabolic pathways of glycine, serine and threonine metabolism were changed for the U251 cell in the treatment with CD9 and CD10 (Figure 5C,D). The metabolic pathways of valine, leucine and isoleucine biosynthesis were disturbed for the U251 cell in the treatment with CD11 and CD12 (Figure 5E,F). In conclusion, in these altered metabolic pathways, the metabolic pathway of aminoacyl-tRNA biosynthesis was the most frequent for the U251 cell in the different coumarin derivatives, including CD1, CD2, CD3, CD4, CD6, CD9, CD10 and CD12. Then, the metabolic pathway of D-glutamine and D-glutamate metabolism also had with a high rating, including CD2, CD3, CD4, CD5, CD7 and CD10. The remaining significantly disturbed metabolic pathways included glycine, serine and threonine metabolism, taurine and hypotaurine metabolism, arginine biosynthesis, alanine, aspartate and glutamate metabolism, phenylalanine, tyrosine and tryptophan biosynthesis, glutathione metabolism and valine, leucine and isoleucine biosynthesis. The results of significantly disturbed metabolic pathways for each coumarin derivative affecting the cellular U251metabolic pathways are shown in Appendix A. Due to the structural similarity of CDs, there was a partial agreement in the metabolic pathways that were perturbed in U251 cells.

## 4. Discussion

In the past few decades, the study of selective MAO-B inhibitors has been continually increasing because of their critical role in regulating synaptic function and monoamine metabolism [47]. However, we were still far from fully understanding the biological processes related to the role of MAO-B inhibitors in the treatment of AD. Tiziani et al. used NMR-based metabolomics to screen a kinase inhibitor library [13]. Similar strategies have been used to personalize treatment [48]. In this study, we collected characteristic metabolites related to neuronal cells in MAO-B inhibitor conditions by using NMR-based metabolomics and associated multivariate statistical analysis. We systematically explored these MAO-B inhibitors’ effects on the metabolic phenotype of neuronal cells to describe the relevant biochemical processes.

According to metabolic pathway analysis, we found that the aminoacyl-tRNA biosynthesis was the most frequent when the U251 cell was in the different MAO-B inhibitors. Aminoacyl-tRNA biosynthesis is the core metabolic pathway of organisms, and the aminoacyl-tRNA synthetases (ARSs) are universally expressed enzymes accountable for charging tRNAs with their cognate amino acids, which is crucial for the first step of protein synthesis [49,50]. There have been many studies showing that ARSs are implicated in some form of neurological disorders, including AD [51]. This result indicated that our coumarin derivatives could be used for AD treatment by regulating this metabolic pathway. The metabolic pathway of D-glutamine and D-glutamate metabolism was also regulated in the U251 by treatment with different coumarin derivatives. Glutamate is the brain’s main excitatory neurotransmitter. The metabolic pathway of the D-glutamine and D-glutamate metabolism could balance the crucial amino acid between neurons and astrocytes [52,53]. Busche et al. found that abnormal glutamate signaling appears in the early stages of AD pathology [54,55]. In our study, the coumarin derivatives that we synthesized could affect the metabolic pathway of D-glutamine and D-glutamate metabolism and thus play a role in the treatment of AD. Meanwhile, our coumarin derivatives could alter glycine, serine and threonine metabolism. The metabolic pathway of glycine, serine and threonine metabolism is the core of the one-carbon metabolism. The metabolic pathway of glycine, serine and threonine metabolism has been reported to be involved in hydrogen sulfide (H2S) metabolism, which was shown to be a neuromodulator in humans [56]. Loïc Dayon et al. observed significant improvements in the prediction of cognitive impairment by adding one-carbon metabolites, which were discovered by mass spectrometry-based metabolomics studies in cerebrospinal fluid and plasma collected from older community-dwelling adults with cognitive impairment, and corresponding asymptomatic volunteers [57]. The results of these studies suggested that our coumarin derivatives could regulate cognitive function. In addition, there were four metabolic pathways participating in one-carbon metabolism, including arginine biosynthesis, alanine, aspartate and glutamate metabolism, phenylalanine, tyrosine and tryptophan biosynthesis and valine, leucine and isoleucine biosynthesis. At the same time, these four metabolic pathways were also involved in the amino acid metabolism. Tynkkynen, J et al. discovered that the amino acid metabolism was related to Alzheimer’s disease by using metabolomics in eight prospective cohorts with 22,623 participants [58]. These perturbed metabolic pathways in our study were also involved in the regulation of the APOE gene [59], which was the strongest genetic risk factor for AD [60,61].

In addition, two metabolic pathways were perturbed for the U251 cell in the treatment of coumarin derivatives, including taurine and hypo-taurine metabolism and glutathione metabolism. These two perturbed metabolic pathways were involved with oxidative stress. Huang et al. found that the perturbed metabolic pathway of taurine and hypo-taurine metabolism could lead to oxidative stress injury in neurons, and worsening dementia by using metabolomics [62]. The glutathione metabolism may have partial neuroprotective effects by reducing mitophagy-related oxidative stress and maintaining mitochondrial function through its effect on autophagy [63]. These results suggested that these coumarin derivatives synthesized by us could affect the development of AD through metabolic pathways involved in oxidative stress.

## 5. Conclusions

The described pathways are linked to nerve cell function related to aminoacyl-tRNA biosynthesis, glutamine and glutamate metabolism, one-carbon metabolism and oxidative stress. Compared with traditional drug screening, the NMR-based metabolomics support the findings of the metabolic pathways associated with drug action. Our study reported the effect of our coumarin derivatives on nerve cells in vitro through their effect on the metabolic phenotype. We believe that these NMR-based metabolomics might accelerate the process of drug research in vitro and in vivo.

## Figures and Tables

**Figure 1 biomolecules-13-00600-f001:**
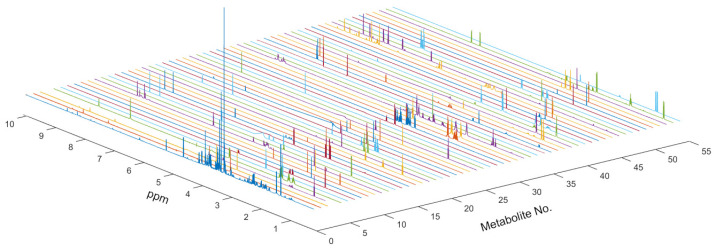
NMR spectrum of metabolites used on AQuA of the global spectrum. Fifty-eight metabolites used in this AQuA were confirmed by using HMDB and BMRB data.

**Figure 2 biomolecules-13-00600-f002:**
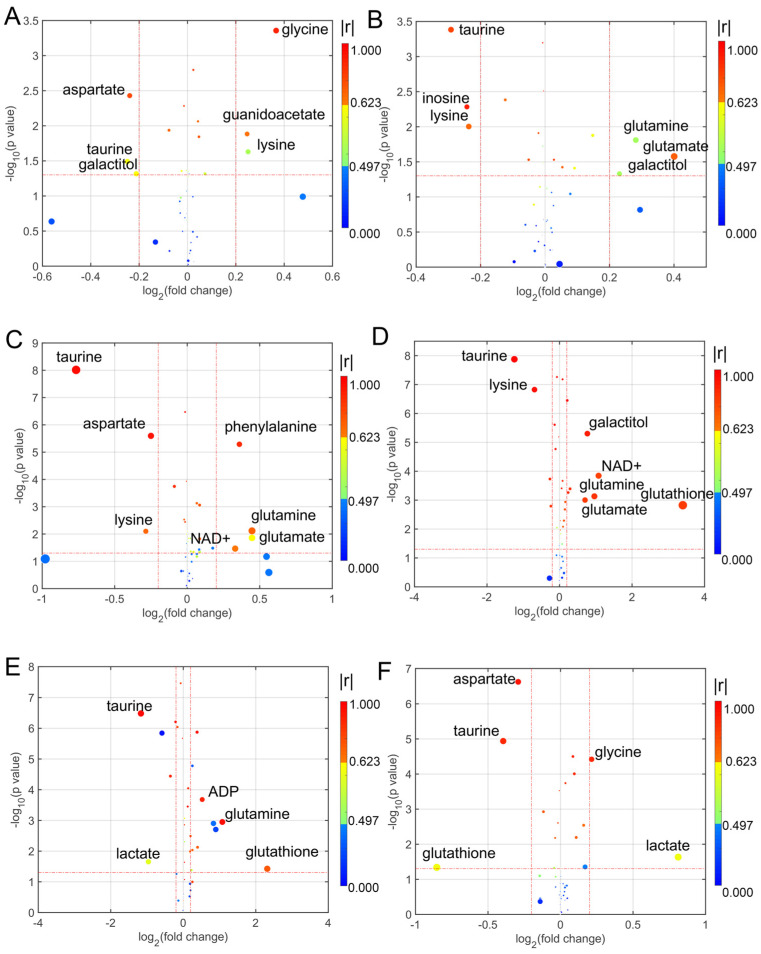
Enhanced volcano plots showing significantly different metabolites. (**A**) CD1 group vs. Control group; (**B**) CD2 group vs. Control group; (**C**) CD3 group vs. Control group; (**D**) CD4 group vs. Control group; (**E**) CD5 group vs. Control group; (**F**) CD6 group vs. Control group. Volcano plots show log_2_ (fold change) on the *x*-axis and −log_10_ (*p* value) on the *y*-axis. Each point represents a metabolite. The circles’ size and color are determined based on the variable importance projection (VIP) and absolute correlation coefficient values (|r|), respectively. For each comparison, the larger the VIP value the larger the size of the circle, and the warmer color corresponds to higher |r|. The gradient blue means |r| is less than 0.497; the gradual bright yellow means |r| is greater than 0.497 and is less than 0.623; the gradient red means |r| is greater than 0.623.

**Figure 3 biomolecules-13-00600-f003:**
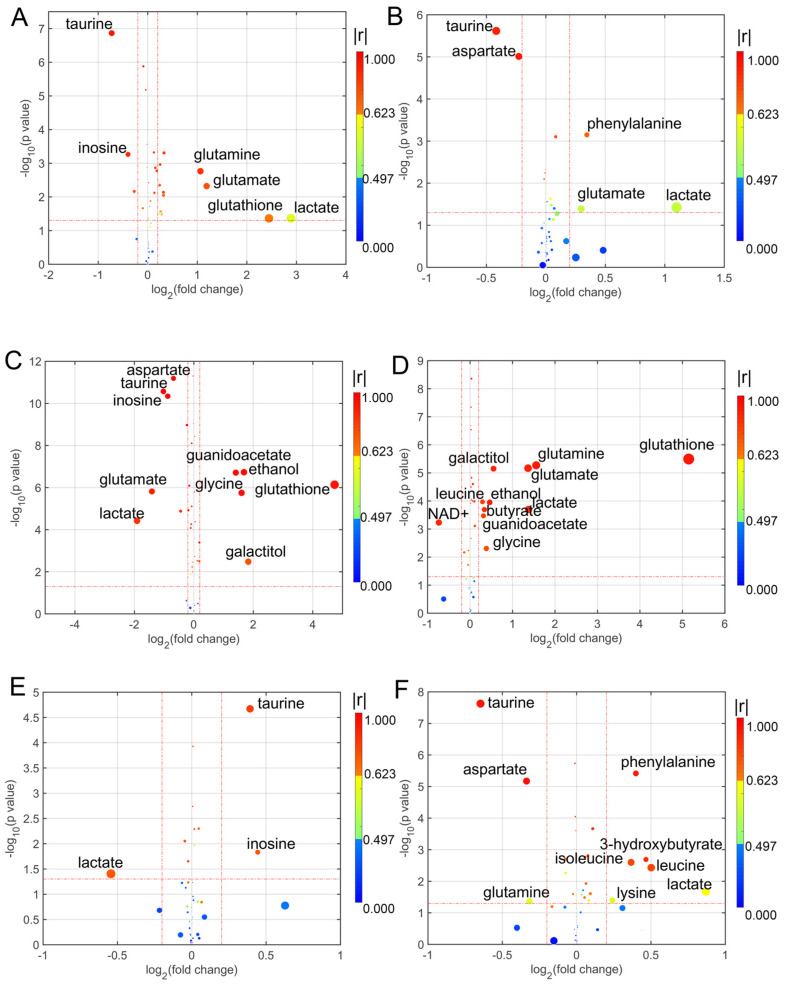
Enhanced volcano plots showing significantly different metabolites. (**A**) CD7 group vs. Control group; (**B**) CD8 group vs. Control group; (**C**) CD9 group vs. Control group; (**D**) CD10 group vs. Control group; (**E**) CD11 group vs. Control group; (**F**) CD12 group vs. Control group. Volcano plots show log2 (fold change) on the *x*-axis and −log10 (*p* value) on the *y*-axis. Each point represents a metabolite. The circles’ size and color are determined based on the variable importance projection (VIP) and absolute correlation coefficient values (|r|), respectively. For each comparison, the larger the VIP value the larger the size of the circle, and the warmer color corresponds to higher |r|; the gradient blue means |r| is less than 0.497; the gradual bright yellow means |r| is greater than 0.497 and is less than 0.623; the gradient red means |r| is greater than 0.623.

**Figure 4 biomolecules-13-00600-f004:**
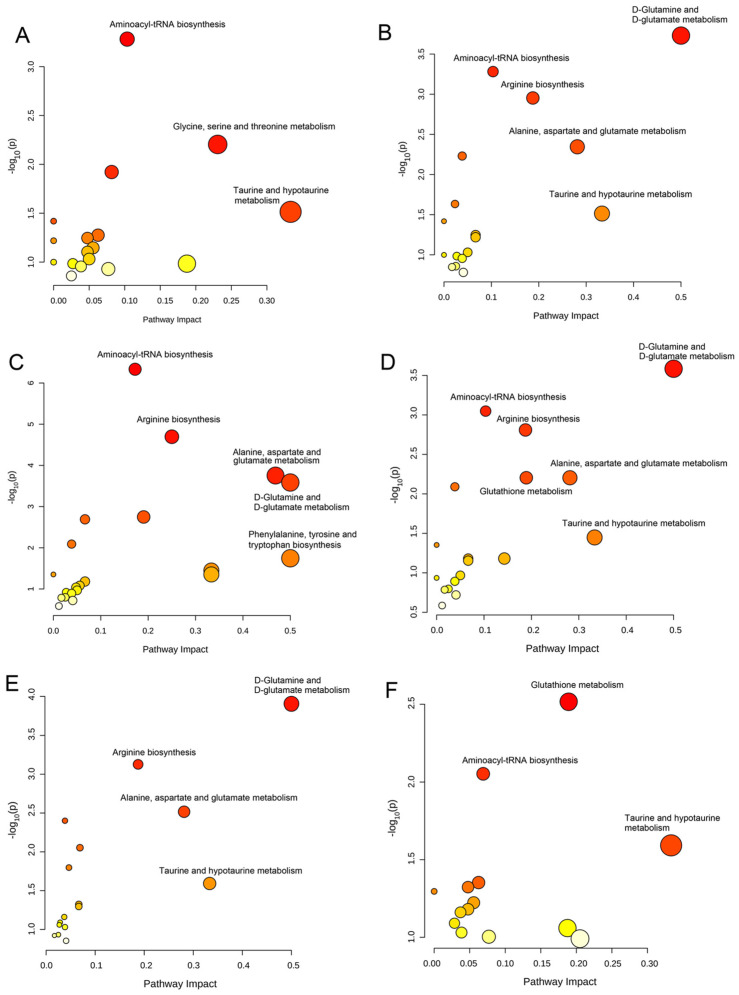
Significantly disturbed metabolic pathways were calculated in comparison between different coumarin derivative groups and the control group. (**A**) CD1 group vs. Control group; (**B**) CD2 group vs. Control group; (**C**) CD3 group vs. Control group; (**D**) CD4 group vs. Control group; (**E**) CD5 group vs. Control group; (**F**) CD6 group vs. Control group.

**Figure 5 biomolecules-13-00600-f005:**
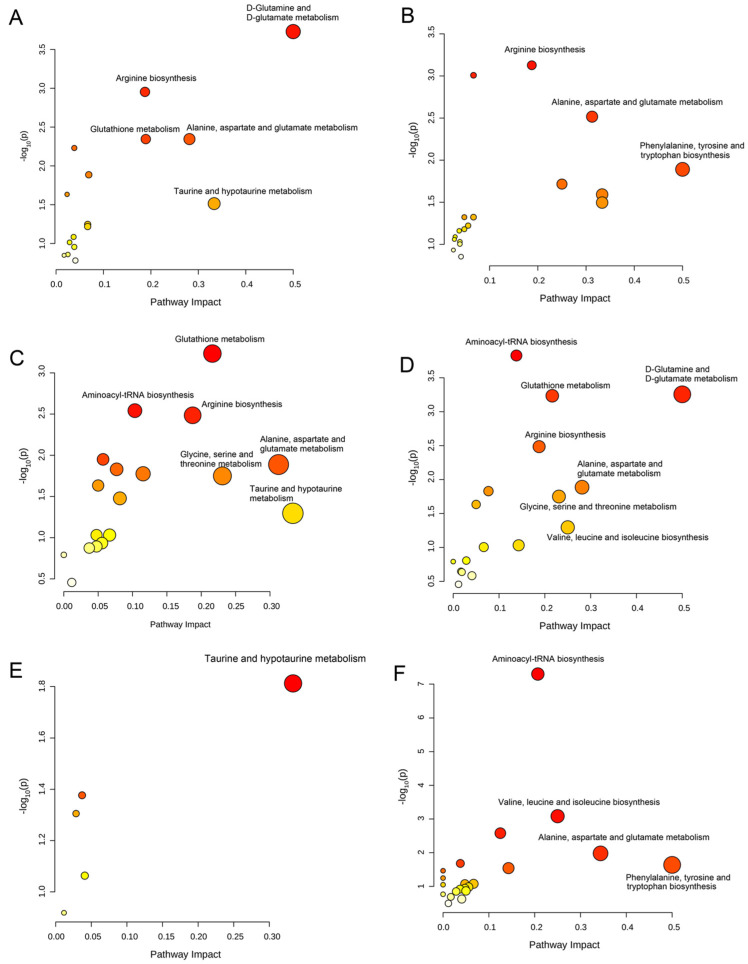
Significantly disturbed metabolic pathways were calculated in comparison between different coumarin derivative groups and the control group. (**A**) CD7 group vs. Control group; (**B**) CD8 group vs. Control group; (**C**) CD9 group vs. Control group; (**D**) CD10 group vs. Control group; (**E**) CD11 group vs. Control group; (**F**) CD12 group vs. Control group.

## Data Availability

The original contributions presented in the study are included in the article/Appendix A; further inquiries can be directed to the corresponding author.

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
