# Peer review of "NMR-Based Metabolomics to Analyze the Effects of a Series of Monoamine Oxidases-B Inhibitors on U251 Cells"

_biomolecules, 2023, doi:10.3390/biom13040600_

Round 1

Reviewer 1 Report

Manuscript has been well written, systematic, and very concise manner, with excellent scientific evidence.

Few shortcomings has been observed, which needs author attention:

·         Detailing about metabolic pathway would add little more value.

·         Coumarin derivative info is missing in main text paper, but only written in abstract.  Please include in the introduction or material and method section. Graphical representation will be encouraged.

·         Line no. 83, 90 : Experiment procedure should be detailed in the respective section along with citing  a reference. This always help reader to connect with the manuscript.

Author Response

Response to Reviewer 1 Comments

General comments:

Manuscript has been well written, systematic, and very concise manner, with excellent scientific evidence.

Response: Thank you very much for your comments, and we will modify them in the subsequent manuscript.

Point-by-point Responses to the Reviewer’s Comments

Point 1: Detailing about metabolic pathway would add little more value.

Response 1: Thanks this comment. We added the meaning of changes in metabolic pathways in the Results and Discussion sections. The relevant modifications were revised in the paper and marked in bright yellow.

Point 2: Coumarin derivative info is missing in main text paper, but only written in abstract. Please include in the introduction or material and method section. Graphical representation will be encouraged.

Response 2: Thanks this comment. The coumarin derivative were synthesized and purified in our Lab. The related synthesis process has been published in JOURNAL OF ENZYME INHIBITION AND MEDICINAL CHEMISTRY (https://doi.org/10.1080/14756366.2019.1634703). The graphical representation has been in this literature. We added this literature in the revised manuscript.

Point 3: Line no. 83, 90 : Experiment procedure should be detailed in the respective section along with citing  a reference. This always help reader to connect with the manuscript.

Response 3: Thanks this comment. According to your suggestion, we added detailed procedure description in the respective section.

Reviewer 2 Report

Comments are in the attached document.

Author Response

Response to Reviewer 2 Comments

General comments:

I am not certain I understand what the main idea of the work is, but I assume that the authors set out to develop a metabolomic approach based on NMR spectroscopy to study the efficiency of MAO inhibitors based on coumarin derivatives and the intracellular metabolic changes they induce. Also, more recent and relevant references should be cited by the authors (more examples below). The manuscript is difficult to follow due to insufficient English language, which should be improved substantially. However, the topic and the experimental design seem to be relevant, therefore, the manuscript could become suitable for publication in Biomolecules, but only after a major review.

Response: Thank you very much for your comments, and we will modify them in the subsequent manuscript. Our goal was to study the pharmacodynamics of coumarin derivatives as MAO inhibitors by using NMR-based metabolomics. In the revised manuscript, we polished the language and added recent and relevant references.

Point-by-point Responses to the Reviewer’s Comments

Point 1: There seems to be something grammatically incorrect about the title. It should be revised.

Response 1: Thanks the reviewer’s comments. We had changed the title in the revised manuscript.

Point 2: The abstract should be re-written to highlight what was done in the current study and what the major advancements are.

Response 2: Thanks the reviewer’s comments. We had re-written the abstract in the revised manuscript to highlight what was done in the current study and what the major advancements are.

Point 3: Line 39: the authors should also include in silico tests, which offer the highest throughput.

Response 3: Thanks the reviewer’s comments. We had added silico tests and relevant reference into our revised manuscript.

Point 4: Line 59: recent publications on the subject should be discussed here since MAO is inhibition is directly related to the main subject of the paper (e.g. 10.1016/j.molstruc.2022.134138[1]; 10.3390/chemistry3030069[2]; 10.3390/MOLECULES25245908[3]; 10.1021/acsomega.2c06906[4], etc.)

Response 4: Thanks the reviewer’s comments. According to your suggestion, we have added these four literatures in this section and illustrated them. Relevant statements are highlighted throughout this article.

Point 5: Also, the properties and mechanisms of known successful inhibitors should be briefly discussed in the introduction, since the mechanism of reversible/irreversible inhibition and catalytic mechanism have now been established (e.g. 10.3389/fchem.2018.00169[5]; 10.1002/anie.201706072[6], 10.4155/fmc.14.23[7], etc.)

Response 5: Thanks the reviewer’s comments. According to your suggestion, we have added these literatures in this section and illustrated them. Relevant statements are highlighted throughout this article.

Point 6: Line 93: so the analytes of interest were left in chloroform after evaporation of MeOH:H2O? It should be clarified. Also, by which means was the membrane of the cells disrupted? Only by the solvent?

Response 6: Thanks the reviewer’s comments. The purpose of our sample pretreatment was to extract the polar metabolites in accordance with references. The polar metabolites were in the MeOH/H2O, and the nonpolar metabolites were in the CHCl3. There were 1x107 cell samples in each group. Relevant statements are highlighted throughout this article.

Point 7: Figure 1 should be moved to Supplementary material.

Response 7: Thanks the reviewer’s comments. We removed Figure 1 into the Supplementary material.

Point 8: Control data seem to be substantially scattered relative to the samples, whereas I would expect it to be more tightly grouped? The authors should elaborate on that because it would seem that coumarin derivatives have no significant influence on the cell’s metabolism

Response 8: Thanks the reviewer’s comments. Results showed that the control samples seem to be substantially scattered relative to the other samples. The other groups of samples were affected by coumarin derivatives and therefore had a more concentrated metabolic phenotype than the control samples. From this point of view, we suggest that the metabolic phenotype is influenced by coumarin derivatives.

Point 9: I propose the figures 3 & 4 and figures 5 & 6 to be merged in a suitable and clear manner so that all the data is available in one place.

Response 9: Thanks the reviewer’s comments. That's a good suggestion, but there are so many groups that we can't merge the figures 3 & 4 and figures 5 & 6 together. If taken together, the picture does not clearly show the differential metabolites and perturbed metabolic pathways of each group.

Point 10: How do the authors know that coumarin derivatives target MAO and not (also) other enzymes? Namely, different enzymes are involved in the pathways the authors described. Authors should elaborate in text.

Response 10: Thanks the reviewer’s comments. These coumarin derivatives were synthesized as MAO-B inhibitors. Related work has been published in JOURNAL OF ENZYME INHIBITION AND MEDICINAL CHEMISTRY (https://doi.org/10.1080/14756366.2019.1634703). We added this literature in the revised manuscript.

Point 11: Line 213: authors claim that all coumarin derivatives disturb the metabolic pathways in the same way. Can we therefore conclude that it is not the coumarin moiety, which is involved in the mechanisms described? It should be elaborated in text.

Response 11: Thanks the reviewer’s comments. In the Line 213, what we describe in our paper is that some of the perturbed metabolic pathways are consistent, not all. This phenomenon may be due to the fact that some of these coumarin derivatives have the same structural formula. In our revised version, we have added the relevant sections and highlighted them.

Point 12: Can the authors perhaps elaborate as to how the studied compounds perturb the cell’s metabolism? Which part of the compound(s) could be responsible for the observed changes?

Response 12: Thanks the reviewer’s comments. In this work, the perturb metabolic pathways were calculated by using the differential metabolites when the U251 cells in the treatment with different CDs. Indeed, our work does not directly analyze the effects of CDs on cellular metabolic pathways.

Point 13: There are also other useful materials you could refer to:

- DOI: 10.1021/acs.jpcb.0c06502

-     DOI: 10.1021/acschemneuro.9b00147

- DOI: 10.1002/pca.3180

- DOI: 10.1007/978-1-0716-2643-6_15

-     etc., etc.

Response 13: Thanks the reviewer’s comments. We have added relevant references to the paper and highlighted them in the revision.

Reviewer 3 Report

Dear Authors 

After thoroughly reading your manuscript, I find it very interesting and provides quality information in this field. I also need to highlight the following points: 

1. The Introduction, Materials and methods as well as for the Discussion need extensive English revision. 

2. In the materials and method section, it is written that (lines 71-72) your team prepared the coumarin derivatives. Nothing is mentioned about their preparation, neither in the the methods nor in the discussions. If these compounds are totally new, procedure for their preparation is needed, otherwise, you should give references on how they were prepared. 

3. In the same topic, and in Table S1, the word "detail" should be "detailed" in the caption. Additionally, all prepared compounds have numbers (41, 42, .....), I wonder from where are these numbers, since there is no scheme for their preparation. Also, please be consistent in recording the melting points of the compounds (example compound "42"- rewrite the range so as to become as the others).  

4. Lines 50-51 should be at the end of the Introduction. 

5. Please be consistent in using IC50 vs IC50

6. The Conclusion is not as required. It should contain all results without references (more should be added). 

Regards

Author Response

Response to Reviewer 3 Comments

General comments:

After thoroughly reading your manuscript, I find it very interesting and provides quality information in this field. I also need to highlight the following points

Response: Thank you very much for your comments, and we will modify them in the subsequent manuscript.

Point-by-point Responses to the Reviewer’s Comments

Point 1: The Introduction, Materials and methods as well as for the Discussion need extensive English revision.

Response 1: Thanks this comment. We havd revised the whole manuscript, especially the introduction, materials and methods and discussion part.

Point 2: In the materials and method section, it is written that (lines 71-72) your team prepared the coumarin derivatives. Nothing is mentioned about their preparation, neither in the methods nor in the discussions. If these compounds are totally new, procedure for their preparation is needed, otherwise, you should give references on how they were prepared.

Response 2: Thanks this comment. The coumarin derivative were synthesized and purified in our Lab. The related synthesis process has been published in JOURNAL OF ENZYME INHIBITION AND MEDICINAL CHEMISTRY (https://doi.org/10.1080/14756366.2019.1634703). We added this literature in the revised manuscript.

Point 3: In the same topic, and in Table S1, the word "detail" should be "detailed" in the caption. Additionally, all prepared compounds have numbers (41, 42, .....), I wonder from where are these numbers, since there is no scheme for their preparation. Also, please be consistent in recording the melting points of the compounds (example compound "42"- rewrite the range so as to become as the others).

Response 3: Thanks this comment. We have corrected the relevant errors you raised in the revised draft. The numbers (example “41, 42” et al.) are just our internal numbers and we have deleted them in the supplementary materials.

Point 4: Lines 50-51 should be at the end of the Introduction.

Response 4: Thanks this comment. We made changes in the revised draft.

Point 5: Please be consistent in using IC50 vs IC50

Response 5: Thanks this comment. We used IC50 in the revised draft.

Point 6: The Conclusion is not as required. It should contain all results without references (more should be added).

Response 6: Thanks this comment. We had revised the conclusions in the revised draft.

Reviewer 4 Report

In the manuscript by Guo et al, the authors describe the changes in the metabolic profiles of glioma cell line U251 treated with various coumarin derivatives acting as MAO-B inhibitors using an NMR-based metabolomics approach. The authors identified 58 metabolites, calculated their relative concentrations, and determined the metabolic pathways within the cells that were most affected.

The manuscript topics are interesting and the methods used are state-of-the-art; however, revisions should be made, especially in the chemistry section. Also, the English language is not up to standard and should be improved throughout the manuscript.

1. Title: please change cell to cells and generally reword the (analysis the effects? - unclear)

2. Abstract: it is quite lengthy, please be more concise and mention less unnecessary details.

3. Please add a new table with the structures of the investigated compounds also in the main text and next to it list the determined MTS IC50 values and compare them with MAO-B inhibition (e.g. IC50 values).

4. In the manuscript, the authors assume that their compounds act predominantly as MAO-B inhibitors. However, the coumarin scaffold is well known and can lead to many different pharmacological outcomes. Since the authors study the phenomena at the cellular level, it is difficult to be sure that what they observe in their analysis is acutely a consequence of MAO -B inhibition alone. I think that this aspect and the corresponding caveats should also be widely discussed in the context of the known state of the art.

5. How does MAO-B inhibition (e.g. IC50 values) correlate with what the authors observed at the cellular level. Please provide relevant data and discuss.

6. The result of the study should be compared with similar NMR-based metabolomic studies in the literature to provide a critical assessment of the performed work.

7. The authors synthesized several interesting coumarin compounds. In addition to the characterization data, the authors should also provide a synthetic scheme. All these synthetic data, if they are new, could easily be included in the main text, as they will be of interest to many readers.

8. Some of the references are missing the names of the journals (e.g., refs. 11 and 15)

9. As mentioned above, the English language should be improved throughout the manuscript.

Author Response

Response to Reviewer 4 Comments

General comments:

In the manuscript by Guo et al, the authors describe the changes in the metabolic profiles of glioma cell line U251 treated with various coumarin derivatives acting as MAO-B inhibitors using an NMR-based metabolomics approach. The authors identified 58 metabolites, calculated their relative concentrations, and determined the metabolic pathways within the cells that were most affected.

The manuscript topics are interesting and the methods used are state-of-the-art; however, revisions should be made, especially in the chemistry section. Also, the English language is not up to standard and should be improved throughout the manuscript.

Response: Thank you very much for your comments, and we will modify them in the subsequent manuscript.

Point-by-point Responses to the Reviewer’s Comments

Point 1: Title: please change cell to cells and generally reword the (analysis the effects? - unclear).

Response 1: Thanks this comment. In the revised draft, we have changed the title.

Point 2: Abstract: it is quite lengthy, please be more concise and mention less unnecessary details.

Response 2: Thanks this comment. We have revised the abstract in the revised version to streamline unnecessary descriptions.

Point 3: Please add a new table with the structures of the investigated compounds also in the main text and next to it list the determined MTS IC50 values and compare them with MAO-B inhibition (e.g. IC50 values).

Response 3: Thanks this comment. We added the table into the supplementary material.

Point 4: In the manuscript, the authors assume that their compounds act predominantly as MAO-B inhibitors. However, the coumarin scaffold is well known and can lead to many different pharmacological outcomes. Since the authors study the phenomena at the cellular level, it is difficult to be sure that what they observe in their analysis is acutely a consequence of MAO -B inhibition alone. I think that this aspect and the corresponding caveats should also be widely discussed in the context of the known state of the art.

Response 4: Thanks this comment. That's a good suggestion. Indeed, our work has focused only on the cellular level. Our work is mainly aimed at constructing NMR-based metabolomics for high-throughput screening of potential drugs. This work did not observe the acute consequences of MAO-B inhibition. We will advance the analysis of coumarin derivatives in animal models in our follow-up work

Point 5: How does MAO-B inhibition (e.g. IC50 values) correlate with what the authors observed at the cellular level. Please provide relevant data and discuss.

Response 5: Thanks this comment. We have added the IC50 related assay process and method to the methods section of the revision. And the result of the IC50 was added to the result section.

Point 6: The result of the study should be compared with similar NMR-based metabolomic studies in the literature to provide a critical assessment of the performed work.

Response 6: Thanks this comment. We have added relevant discussions to the revised draft and highlighted them.

Point 7: The authors synthesized several interesting coumarin compounds. In addition to the characterization data, the authors should also provide a synthetic scheme. All these synthetic data, if they are new, could easily be included in the main text, as they will be of interest to many readers.

Response 7: Thanks this comment. The coumarin derivative were synthesized and purified in our Lab. The related synthesis process has been published in JOURNAL OF ENZYME INHIBITION AND MEDICINAL CHEMISTRY (https://doi.org/10.1080/14756366.2019.1634703). We added this literature in the revised manuscript.

Point 8: Some of the references are missing the names of the journals (e.g., refs. 11 and 15)

Response 8: Thanks this comment. In the revised version, we checked the references and completed the incomplete parts.

Point 9: As mentioned above, the English language should be improved throughout the manuscript.

Response 9: Thanks this comment. In the revised version, we checked the references and completed the incomplete parts. We had improved the English language throughout the manuscript.

Round 2

Reviewer 2 Report

The issues raised have been addressed in most part, however, the English language could still be improved. there are multiple typos and grammatical errors present already in the abstract. I recommend a language editing service or the help of a native speaker.

Author Response

Response to Reviewer 2 Comments at Round2

Point-by-point Responses to the Reviewer’s Comments

Point 1: The issues raised have been addressed in most part, however, the English language could still be improved. there are multiple typos and grammatical errors present already in the abstract. I recommend a language editing service or the help of a native speaker.

Response 1: Thanks the reviewer’s comments. Language is really our weakness. We re-revised the manuscript with the help of a native speaker.

Reviewer 4 Report

In the revised version presented, the authors have adequately addressed the comments made.

The authors' response to Point 4 is well formulated and puts the study and its findings in a broader perspective. Therefore, this text should also be included in the discussion or conclusion section of the revised manuscript before it is accepted for publication.

Author Response

Response to Reviewer 4 Comments at Round 2

Point-by-point Responses to the Reviewer’s Comments

Point 1: In the revised version presented, the authors have adequately addressed the comments made.

The authors' response to Point 4 is well formulated and puts the study and its findings in a broader perspective. Therefore, this text should also be included in the discussion or conclusion section of the revised manuscript before it is accepted for publication.

Response 1: Thanks this comment. In the revised manuscript, we have added this section to the discussion section (lines 326-332).